# Comparative Genomics Used to Predict Virulence Factors and Metabolic Genes among *Monilinia* Species

**DOI:** 10.3390/jof7060464

**Published:** 2021-06-08

**Authors:** Marina Marcet-Houben, Maria Villarino, Laura Vilanova, Antonieta De Cal, Jan A. L. van Kan, Josep Usall, Toni Gabaldón, Rosario Torres

**Affiliations:** 1Comparative Genomics Group Barcelona Supercomputing Centre (BSC-CNS), Institute for Research in Biomedicine (IRB), Jordi Girona, 29, 08034 Barcelona, Spain; marina.marcet@bsc.es; 2Department of Plant Protection, INIA, Ctra. de La Coruña Km. 7, 28040 Madrid, Spain; villarino.maria@inia.es (M.V.); cal@inia.es (A.D.C.); 3Postharvest Programme, Edifici Fruitcentre, Parc Científic i Tecnològic Agroalimentari de Lleida, IRTA, Parc de Gardeny, 25003 Lleida, Spain; laura.vilanova@irta.cat (L.V.); Josep.Usall@irta.cat (J.U.); 4Laboratory of Phytopathology, Wageningen University, Droevendaalsesteeg 1, 6708PB Wageningen, The Netherlands; jan.vankan@wur.nl; 5Catalan Institution for Research and Advanced Studies (ICREA), 08010 Barcelona, Spain

**Keywords:** brown rot, secreted proteins, CAZome, secondary metabolism, sexual reproduction

## Abstract

Brown rot, caused by *Monilinia* spp., is among the most important diseases in stone fruits, and some pome fruits (mainly apples). This disease is responsible for significant yield losses, particularly in stone fruits, when weather conditions favorable for disease development appear. To achieve future sustainable strategies to control brown rot on fruit, one potential approach will be to characterize genomic variation among *Monilinia* spp. to define, among others, the capacity to infect fruit in this genus. In the present work, we performed genomic and phylogenomic comparisons of five *Monilinia* species and inferred differences in numbers of secreted proteins, including CAZy proteins and other proteins important for virulence. Duplications specific to *Monilinia* were sparse and, overall, more genes have been lost than gained. Among *Monilinia* spp., low variability in the CAZome was observed. Interestingly, we identified several secondary metabolism clusters based on similarity to known clusters, and among them was a cluster with homology to pyriculol that could be responsible for the synthesis of chloromonilicin. Furthermore, we compared sequences of all strains available from NCBI of these species to assess their MAT loci and heterokaryon compatibility systems. Our comparative analyses provide the basis for future studies into understanding how these genomic differences underlie common or differential abilities to interact with the host plant.

## 1. Introduction

*Monilinia* spp., necrotrophic Ascomycetes of the family *Sclerotiniaceae*, are the main pathogens responsible for brown rot in stone and some pome fruits. They are present in numerous fruit areas of the world, and are responsible for important economic losses in the pre-harvest and the post-harvest periods [1]. Brown rot can appear at any stage of fruit development, during and after harvest [2], and infect fruit either through an opening (micro-cracks, lenticels, wounds, or stomata) or through contact with an intact surface [3]. Post-harvest losses are usually more severe than pre-harvest losses, and typically occur during storage and transport, in some cases even affecting fruit at the processing stage [4]. The incidence and development of this disease are clearly affected by climatic conditions, and can cause up to 80% fruit loss under favorable environmental conditions [5].

In Europe, brown rot is caused by four species of *Monilinia*: *Monilinia laxa* (Aderh. & Ruhland) Honey, *M. fructicola* (G. Wint.) Honey, *M. fructigena* Honey [1], which is more common in pome than in stone fruit, and, to a lesser extent, *M. polystroma* G. Leeuwen [6]. Among these species, *M. fructicola* is the most virulent, causing larger fruit lesions and displaying shorter incubation and latency periods [7]. *M. fructicola* was classified as an EPPO A2 quarantine pest because it has only been identified in a few EPPO member countries [8]. The relative occurrence of this species has increased over the years to the point that it coexists at the same level with *M. laxa*, and it has partially displaced *M. fructigena* as a pathogen in stone fruit [9,10].

Mature nectarine tissues with visible *M. fructicola* infections are characterized by extensive colonization of the deep subdermal tissues by pathogen hyphae [11]. Fruit tissues and cells are colonized inter- and intracellularly, and this colonization is accompanied by increasing degradation of the cell walls, which is typical in colonization by necrotrophic fungi [11]. Similar to other necrotrophic fungi, *Monilinia* spp. use cell wall degrading enzymes for penetrating the fruit surface and invading and colonizing the fruit [1]. These extracellular activities play an important role in *Monilinia* pathogenesis [1,12]. For many fungal pathogens, some of these degrading enzymes are virulence factors [13]. Cell wall degrading activities by *M. laxa*, *M. fructicola*, and *M. fructigena* have been detected in culture media [14]. *M. fructicola* is known to secrete several enzymes, which can hydrolyze pectins, namely endopolygalacturonases, such as MfPG1 [15], and pectin esterases [16], as well as MfCUT1, which is the main cutinase [17]. *M. fructigena* is known to secrete several enzymes with the same activity as MfPG1 [18] or pectin lyase activity with the ability to macerate apple tissue [19], proteases, and amylases [18]. Enzymes such as catalases, peroxidases, glutamic dehydrogenases, esterases, and alkaline phosphatases have also been reported in *M. laxa* [20]. Additionally, cutinase activity has been reported in *M. laxa* cultures [14]. Cellulase secretion was detected in *M. laxa*, *M. fructigena*, and *M. fructicola* to different extents in different conditions [1]. CAZymes were identified in *M. laxa* exoproteomes in media containing glucose and pectin as sole carbon sources [21].

The future for brown rot control and marketing entails an additional effort to understand the fruit-pathogen interaction to develop new control strategies. The availability of genomes for the main *Monilinia* spp. on stone fruit [22,23,24] coupled with a comparative genome analysis can be useful to elucidate common and different genomic features among organisms, identifying key genes responsible for the host-specificity of each *Monilinia* species. It can also explain the differential ecological fitness of each *Monilinia* spp. in various environments and their rate of spread in different geographical areas. For these reasons, this study outlines and compares genome sequences for European strains of *Monilinia* spp. to conduct a phylome analysis. Furthermore, a comparison was performed among *Monilinia* spp. of several functionally annotated features such as CAZymes, effector proteins, secondary metabolism (SM) gene clusters, the mating locus, and genes involved in pathogenicity.

## 2. Materials and Methods

### 2.1. Sequences and Annotations

Genomes, proteomes and sequencing reads were downloaded for all *Monilinia* species (*M. fructicola*, *M. laxa*, *M. fructigena*, *M. polystroma*, and *M. aucupariae*) and strains available at the NCBI database. In addition, genomes and proteomes were downloaded for ten additional species of the *Leotiomycetes* clade (*Myriosclerotinia scirpicola, Myriosclerotinia curreyana*, *Botrytis tulipae*, *Myriosclerotinia duriaeana*, *Botrytis hyacinthi*, *Botrytis paeoniae*, *Botrytis cinerea*, *Ciborinia camelliae*, *Ciboria shiraiana*, and *Hymenotorrendiella dingleyae*) (see Appendix A). For genomes lacking a gene prediction we used AUGUSTUS v3.3 [25] to predict the location of genes using the training parameters of *Botrytis cinerea*.

The *Monilinia* strains used for phylome reconstruction were *M. fructicola* CPMC6 [23], *M. laxa* 8L [22], *M. fructigena* gena6, *M. polystroma* SP-61, and *M. aucupariae* LMK733. Details can be found in Appendix A.

To ensure the quality of all proteomes used, we assessed the completeness of the genome using BUSCO (v4.0.2) [26] with the helotiales_odb10 dataset. All proteomes with the exception of two of the strains not used in the phylome had a completeness above 90% (see Appendix A).

### 2.2. Phylome Reconstruction

To reconstruct the phylomes, i.e., the complete collection of phylogenetic trees reconstructed for each gene encoded in a genome, the proteomes of 15 *Leotiomycetes* species were collected (see Appendix A). A phylome was reconstructed for each of the five *Monilinia* species present in the dataset, using a slightly modified PhylomeDB pipeline [27]. In brief, for each gene in the genomes of a *Monilinia* species, a BlastP v2.5.0+ [28] search was performed to obtain the list of homologous proteins. Blast results were filtered so that only results that had an e-value below 1 × 10^−5^ and an overlap of more than 50% over the query sequence were kept. The top 150 hits per query protein were kept. For each resulting set of homologous proteins, six multiple sequence alignments were reconstructed using three different programs: MUSCLE v3.8.1551 [29], MAFFT v7.407 [30], and KALIGN v2.04 [31] in both forward and reverse directions [32]. A consensus alignment was then obtained using MCOFFEE, from the tcoffee package v12.00 [33]. This alignment was then trimmed to remove columns that were not supported by the individual alignments (-ct 0.16667) and to remove columns with a large percentage of gaps (-gt 0.1) using TrimAl v1.4. rev15 [34]. The trimmed alignment was used to reconstruct a maximum likelihood phylogenetic tree using IQTREE v1.6.9 [35]. Model selection was performed by IQTREE limiting the choices to 5 of the most common models (DCmut, JTTDCMut, LG, WAG, VT). 1000 rapid bootstraps were calculated per tree. Trees and alignments were uploaded to phylomedb [36] under the phylome IDs 383 to 387 (http://phylomedb.org/, accessed on 31 May 2021).

### 2.3. Phylome Analysis

Each phylome was computationally scanned to determine orthology and paralogy relationships across species using the species overlap algorithm [37] as implemented in ETE3 v3.1.1 [38]. Shortly, for each tree the lineage spanning from the seed protein (protein which initiates the blastP search during the phylome reconstruction, see above) to the root was scanned, and each node was assessed to qualify whether it was a duplication or a speciation. A duplication node was called when leaves at both sides of the node had at least one species in common. Leaves connected through a speciation node were considered orthologs whereas leaves connected through a duplication node were considered paralogs. Expansions were considered when multiple duplication events occurred specifically in one single species of *Monilinia* leading to the presence of at least 5 in-paralogous sequences. As the presence of transposable elements can impact the calculation of such expansions, we performed this analysis after removing trees that contained proteins that had Pfam domains associated with transposable elements. In addition, we also removed any duplication that was suspected to have been called due to an erroneous gene prediction that resulted from protein fragmentation. This was done by scanning the blastP results and observing whether two paralogs were consistently mapping to two parts of the same orthologous protein.

Gene gain and loss was derived from orthology information. So, a gene family was considered to have been gained at the common ancestor of all the species that contained an orthologous gene from the family. Once the gain was inferred, losses were calculated by first marking all the species that had lost the gene family and then minimizing the number of inferred losses by assigning the loss to the common ancestor of monophyletic species that lack the gene. Note that gains can only be assigned to the lineages leading to seed species of the phylome. To obtain a more complete view of gains and losses, we joined calculations from the five computed *Monilinia* phylomes, and for each node the average number of gains or losses was calculated. Proteins were assigned as “orphans” when they had no homologous proteins among the species used in the phylome reconstruction.

### 2.4. Species Tree Reconstruction

Duptree v1.48 [39] was used to reconstruct species trees for each of the phylomes using all the gene trees reconstructed in each phylome. Additionally, a maximum likelihood tree was reconstructed using a concatenation approach based on the phylome of *M. fructicola*. First, 2871 single copy alignments obtained in the phylome were concatenated to obtain a single multiple sequence alignment that contained nearly two million positions (1,992,645). Then, IQTREE v1.6.9 [35] was used to reconstruct the species tree using LG as the model. The rest of the parameters were inferred by IQTREE, and the best model according to the BIC criterion was (LG + F + R4). 1000 rapid bootstraps were calculated. Bootstrap support was 100 except for two nodes (see Figure 1).

### 2.5. Functional Prediction

All proteins used in the phylome were annotated using interproscan v5 [40]. In addition, CAZy proteins were annotated using dbcan metaserver [41]. Predicted CAZy proteins were filtered to retain only those that were annotated by at least two of the three programs in dbcan metaserver. A blastP search was run against the MycoCLAP [42] database (accessed June 2020) using the predicted CAZy proteins as queries. Secreted proteins were predicted using SignalP v5.0. Proteins with a signal peptide were scanned for the presence of transmembrane domains using TMHMM. Proteins with multiple TMHMM domains or with a single TMHMM domain that fell outside the first 60 AA of the protein were omitted. EffectorP 2.0 [43] was then used to detect effector genes in the predicted secretomes of *Monilinia* species. GO term enrichment was calculated using an in-house adaptation of FatiGO [44].

Secondary metabolism (SM) gene clusters were detected using the SMURF webserver [45], accessed June 2020. Proteomes were also scanned for the presence of known SM gene clusters by performing a blastP search between a collection of 174 gene clusters obtained from the literature and the proteomes included in the phylome. The presence of a cluster was assumed when a large part of the cluster was present in a contiguous region of a genome in one of the species. For short clusters (i.e., having 4 or fewer genes), all genes were required to be present. For longer clusters, at least half of the genes were required to be present as long as at least 3 genes were found in a cluster. A blastP search against the NCBI nr database was performed for all orphan genes in order to see whether they had homologs outside the phylome. Only strains of *Monilinia* included in the phylome underwent this thorough analysis (*M. fructicola* CPMC6 [23], *M. laxa* 8L [22], *M. fructigena* gena6, *M. polystroma* SP-61, and *M. aucupariae* LMK733.)

MAT loci were detected by doing a blast search using the MAT loci taken from *B. cinerea* (MAT-1-1-1: AHX22632.1, MAT-1-1-5: AHX22633.1, MAT-1-2-1: AHX22634.1, MAT-1-2-10: AHX22635.1, SLA2: EMR81653.1 and APN2: EMR81656.1). HET proteins were located by scanning the interproscan annotations for the presence of the HET (PF06985) or HET-C (PF07217) domains. This last set of analyses was performed on all strains of *Monilinia* included in this paper.

### 2.6. Monodictyphenone Gene Trees Reconstruction

The cluster of the secondary metabolite monodictyphenone was explored for evidence of horizontal gene transfer. For each protein encoded in the cluster, a blast search was performed against a local database including 340 fungal genomes [46]. The results were added to the ones found in *Monilinia*, and a phylogenetic tree was built using the same methodology as detailed for the phylome reconstruction. ETE3 v3.1.1 [38] was used to explore the resulting topologies and obtain the images.

### 2.7. Strain Comparison

Reads belonging to seven *Monilinia* strains were downloaded from SRA (see Appendix A). Reads were mapped to their own reference genome using BWA mem v0.7.17-r1188 [47] and SNPs were called using GATK v4.0.4.0 [48]. Bedtools v2.29.2 [49] was then used to calculate genome coverages. Missing genes were obtained by checking coverage at their positions. If no reads were mapping at a position, it was considered not present. Genes with less than 90% of their coding position present were considered missing. Note that this approach avoids labeling genes that are not part of the gene catalog as a result of annotation errors due to sequencing or assembly errors causing frameshifts or spurious internal stop codons as missing.

### 2.8. Genes Involved in Pathogenicity

Genes related to the early infection process of *M. laxa* strain 8L on nectarines from RNA-Seq analysis [50] were searched for in four different species of *Monilinia* (*M. fructicola* CMPC6 strain, *M. fructigena* gena6 strain, *M. polystroma* SP-61 strain, and *M. aucupariae* LMK733 strain).

## 3. Results and Discussion

### 3.1. Comparison of Monilinia Genomes

Phylomes are complete collections of phylogenetic trees built for each gene encoded in a genome of interest. They allow a broad view of the evolutionary forces that have shaped a species’ genome. We reconstructed five phylomes, one for each sequenced *Monilinia* species: *M. laxa* 8L [22], *M. fructicola* CPMC6 [23], *M. fructigena* gena6, *M. polystroma* SP-61, and *M. aucupariae* LMK733. A total of 15 proteomes were used to reconstruct the phylomes (see Appendix A), including the five *Monilinia* spp. and ten other Leotiomycetes, including the well-annotated necrotroph *B. cinerea* [51,52]. Based on the information of these phylomes, and using a combination of a super-tree and gene concatenation approaches, we reconstructed a species tree representing the evolutionary relationships of the 15 Leotiomycetes species. The five phylomes resulted in the exact same species tree (Figure 1). The resulting topology was well supported and confirmed the monophyly of the five *Monilinia* spp. as compared to the other 10 taxa (Figure 1). *M. aucupariae*, a non-pathogen on stone fruit, was the first diverging species followed by *M. fructicola, M. laxa*, and finally the two species that were until recently considered to be the same species: *M. fructigena* and *M. polystroma* (see Figure 1). These results confirmed those obtained by Villarino et al. [7], where the dendrogram generated based only on the ability to produce sclerotia on potato dextrose agar and lesion length on infected fruit separated *Monilinia* isolates into two clusters. One cluster was composed of *M. laxa* and *M. fructigena* isolates, and the other comprised *M. fructicola* isolates. Species from the *Myriosclerotinia* genus and the *Ciborinia* genus were sisters to the *Monilinia* clade. It is noteworthy that *Myriosclerotinia* species are the only ones in the analysis that are not necrotrophs, but rather biotrophs. Orthology and paralogy relationships were calculated from the phylomes using the species overlap algorithm [37], and the number of gains and losses that had occurred in each node were mapped to the species tree (see Methods, and Figure 1). Of note, a very large number of gene losses was inferred specifically in *M. aucupariae* (1348), despite the fact that this species has the largest genome sequence among *Monilinia* species. This difference cannot be attributed to the quality of the genome or annotation as BUSCO results show similar results for other *Monilinia* species. This wave of gene loss was not associated with enrichment in any gene ontology (GO) terms.

We detected 1021 orphan genes in the five *Monilinia* species that had no homologs in any of the other species used in the phylome. Out of these, 913 did not have a hit in the nr database and may represent either annotation errors or truly species-specific genes, 35 had hits in other Leotiomycetes not included in the phylome, and an additional 24 had hits in other *Pezizomycotina* species. Four proteins had a best hit in other fungal proteins, though their identity was low. A closer look at those four sequences revealed that they were either very short, had low complexity regions, or both. The remaining proteins (44) had hits outside fungi and may represent cases of horizontal gene transfer (HGT) or more likely contamination or miss-annotations.

We examined the phylomes for duplications that happened specifically in the lineage of *Monilinia,* which diverged 14.4 MyA according to a recent study [53]. There were very few duplications in these nodes (Appendix A). This can be partially attributed to the short evolutionary time of *Monilinia* species and partially to a low duplication rate. Of note is a gene family expansion of the phytotoxin *Bcnep1* specific to *M. aucupariae* (see Appendix A). *Bcnep1* and its paralog *Bcnep2* have been described in *B. cinerea* [54], and cause necrosis in dicotyledonous plants. All *Monilinia* species have a single copy of *Bcnep2*, but whereas most have also one single copy of *Bcnep1*, we found up to 20 copies of *Bcnep1* in *M. aucupariae*. These duplicated genes are dispersed throughout the genome and therefore do not result from tandem duplications. Appendix A contains a summary of this and subsequent analyses performed in this study.

### 3.2. Secretome Comparison in Monilinia Species

Necrotrophs secrete a range of cell wall degrading enzymes that serve to break plant cells and acquire nutrients [55]. Secreted proteins, as well as cell wall degrading enzymes, can be regarded as virulence factors, and as such can be related to the pathogenesis of a species. A list of secreted proteins found in the 15 species used in the phylome was obtained using SignalP v5.0 [56] (see methods, Appendix A, and Appendix A).

The number of secreted proteins ranges from 628 to 1,203 in the set of 15 species used in the phylome analysis (see Appendix A). 374 potential effectors were detected, which represented roughly 10% of the secretome. Differences in secretome size between different species are smaller when the number of secreted proteins is normalized by proteome size. On average, secretomes account for roughly 7.5% of a species’ protein content (range 6.56% to 8.29%). In absolute terms, the species with the smallest secretome was *M. aucupariae* (628). We grouped secreted proteins into orthologous groups and searched for families specific to *Monilinia* species. Seventeen families were exclusively present in more than one *Monilinia* species, of which five were present in all five *Monilinia* species.

Secreted proteins that were specific to *Monilinia* species and present in all five species included a glucose-methanol-choline oxidoreductase and two proteins annotated as Berberine-like. These last proteins could be homologous to proteins found in the Pyriculol cluster of *Magnaporthe oryzae* [57]. Pyriculol and its derivatives produce leaf lesions, although the deletion of this cluster in *M. oryzae* did not affect virulence in rice [57]. A homolog of this cluster is present in all five *Monilinia* species (see Figure 2A and Appendix A). One of the proteins in the cluster, shown in Figure 2A marked with an X, was not thought to be involved in the synthesis of pyriculol in the original description of the cluster [57] but is conserved across the *Monilinia* species. The presence of an additional dehydrogenase (Y in Figure 2A) and the conservation of gene X could indicate that the compound synthesized by the *Monilinia* cluster is not Pyriculol but a related compound. Sassa et al. [58] described the presence in *M. fructicola* of monilidiol, a compound with antibiotic and phytotoxic activity similar to pyriculol. It is possible that the compound synthesized by this cluster is monilidiol [58]. Alternatively, the PKS encoded in this cluster matches a fragment sequenced that is thought to be responsible for synthesis of the antibiotic and antifungal chloromonilicin [59]. Further research is necessary to establish whether this cluster is responsible for the synthesis of monolidiol, chloromonicilin, or an alternative compound.

Also specific to *Monilinia*, though not found in all the species, was a peptidase of the subtilase family found in *M. fructigena*, *M. fructicola*, and *M. laxa*. We used EffectorP v2.0 [43] to detect potential effector proteins among the set of secreted proteins in *Monilinia*. Out of a total of 3612 secreted proteins in all five *Monilinia* species, 374 (10.4%) were detected as potential effector proteins (see Appendix A). Using orthologous relationships, we were able to group 330 out of the 374 effector proteins into 127 families, leaving 44 effector proteins unmatched. Only 20 of the families were present in all five *Monilinia* species. Among the proteins detected as effector proteins by EffectorP [43] are the homologs to the phytotoxin BcNEP1.

### 3.3. The CAZome

Another set of potential enzymes that could contribute to the pathogenic capacity of the different species are the CAZy proteins, as they degrade complex carbohydrates. We predicted all CAZy proteins in the species considered (Appendix A and Appendix A). Overall, 6261 proteins out of the 161,183 proteins included in the phylome were annotated as CAZy. Out of the 6261 CAZy proteins in our phylome, 1494 presented hits with at least 50% identity to mycoCLAP proteins. The differences in numbers of CAZy proteins found in the different species is limited (see Figure 1 and Appendix A). In absolute terms, *Botrytis* species and the outgroup *H. dingleyae* have the largest number of CAZy proteins, whereas *Monilinia* and *Myriosclerotinia* have low numbers of CAZy proteins. Number of CAZy proteins range between 373 and 497. When dividing the number of CAZy proteins by proteome size, the species with the lowest percentage of CAZymes in its genome is the outgroup (3.4%). On the other hand, *Monilinia* are all on the upper part of the table (range between 4.1% and 4.2%), with only *Ciboria shiraiana* (4.1%) having a similar percentage of its proteome dedicated to CAZome.

A transcriptomic analysis of *M. fructicola*, *M. laxa*, and *M. fructigena* conducted under in vitro conditions identified carbohydrate-active enzymes [12], but the role of these enzymes in infection of peach fruit requires more detailed transcriptome analyses and functional analysis by gene knock-out mutants. So, we explored the variability of the CAZy proteins in different species by dividing them into different families (see Appendix A). Most families showed little differences in numbers of CAZy family members, and there was no *Monilinia*-specific CAZy family. Therefore, we focused on those families that showed the largest variability in their presence in different species. The largest variability is observed in family AA3, which contains auxiliary proteins that catalyse the oxidation of alcohols or carbohydrates with the formation of hydrogen peroxide or hydroquinones. These proteins have been found in several wood-degrading fungi. While they are present in all *Monilinia* species, they are even more abundant in the three *Botrytis* species that infect non-woody tissues such as leaves, flowers, and fruit.

Another CAZyme family with significant variation is GH28, a family of pectin degrading enzymes that has been reported to be expanded in necrotrophs [60]. The results presented here are consistent with those findings, as *Myriosclerotinia* species, which are biotrophs, have fewer copies (average of 11) than the necrotrophs *Botrytis* and *Monilinia* (averages of 19 and 17, respectively). Most proteins in the GH28 family are polygalacturonases, though some of them hydrolyze rhamnogalacturonan, which forms a branched part of the pectin complex. Within the GH28 family, three orthogroups were specific to necrotrophic species and these could be annotated more precisely based on homology to the MycoCLAP database [42].

Considering these results, differences in CAZy proteins between *Monilinia* and other species seem to be centered mainly around two families, AA3 and GH28, though smaller differences in the number of homologs in other families can be found.

### 3.4. Genes Involved in Pathogenicity

The availability of the genome of *M. laxa* 8L [22] and a study of the transcriptome of this pathogenic species on nectarines previously conducted by Balsells-Llauradó et al. [50] allowed a search for possible genes related to pathogenesis. After analysis of the RNA-Seq results, a set of 22 genes was selected for their expression early in nectarine infection (Appendix A) and their relationship with pathogenesis in *Monilinia* spp. or *B. cinerea* (PHI database [61]) (Appendix A). The transcripts to *M. laxa* were de novo annotated for multiple functional categories, including Gene Ontology (GO) terms, carbohydrate-active enzymes (CAZymes), Pfam, fungal peroxidases (fPox), genes involved in pathogen-host interactions (PHI), membrane transport proteins (TCBD), and proteins with signal peptides (SignalP), among others. The presence of orthologs of these 22 genes was checked in the other four species of *Monilinia* considered. GO terms classification for these genes showed a significant overrepresentation of CAZymes related to plant cell wall degradation (Appendix A). Appendix A includes polygalacturonase 1 (MlPG1), and pectin lyase 2 gene (MlPNL2) described by Rodríguez-Pires et al. [62], as well as genes involved in oxidative-reduction processes and transmembrane transport. All of them had orthologs in all *Monilinia* species with the exception of Phy00AEVLC_61186, which encodes a transmembrane transporter and was missing in *M. aucupariae* (Appendix A).

### 3.5. Secondary Metabolism Clusters

We examined the 15 proteomes used in the phylome for the presence of gene cluster families that were homologous to a collection of 174 known SM clusters (updated from Ballester et al. [63]). A cluster was considered present when most of the cluster genes were present, with the understanding that the synthesized compound may be related but not necessarily the same (see Appendix A and Appendix A). Among the clusters present in *Monilinia*, one was the aforementioned one homologous to the pyriculol gene cluster in *M. oryzae,* and the sordarial gene cluster in *Neurospora crassa* [64]. Another interesting cluster was a homolog of the cluster synthesizing monodictyphenone (see Figure 2B), a prenyl xanthone derivative. Xanthones have pharmacological uses as antifungal and antibacterial compounds and are produced by several groups of plants and fungi [65]. The monodictyphenone gene cluster was described in *Aspergillus nidulans* [66] and consists of 12 genes, two of which (*mdpI* and *mdpD*) are not thought to be involved in the synthesis of Monodictyphenone. A similar cluster is present in four of the five *Monilinia* species, missing only in *M. aucupariae*. The clusters of the four species are identical, with the only difference that *mdpG* in *M. fructigena* is absent in the annotation. This can either be due to a miss-assembly or genuinely be missing from this species in which case the cluster would not produce any compound. When comparing the clusters of *Monilinia* to the one in *A. nidulans*, they share most genes. The few differences include the lack of *mdpI* and *mdpJ*, the duplication of *mdpH*, and the presence of two additional genes (white arrows in Figure 2). Gene order is not conserved between the clusters of *Monilinia* and its homolog in *A. nidulans*. As many SM gene clusters have been transferred between unrelated fungal taxa, we considered the possibility that this was another such case. We searched homologous clusters in a database formed by 340 fungal genomes and obtained a list of 22 clusters from four of the subphyla of Pezizomycotina (Leotiomycetes, Sordariomycetes, Eurotiomycetes and Dothideomycetes) (Appendix A). For each protein family, a gene tree was reconstructed to determine their phylogenetic relation to the clusters in *Monilinia*. All trees followed a similar pattern in which the sequences from *Monilinia* were far related from the ones from *A. nidulans.* The phylogenetic distance, coupled with the difference in protein composition of the gene cluster, could indicate that the MDP cluster in *Monilinia* diverged long ago from the homologous clusters in other taxa, so they share a very ancient ancestor. The *Monilinia* sequences were always grouped with an ortholog in *Sclerotinia borealis*, another member of the Leotiomycetes subphylum that infects barley, rye, and wheat. Orthologs in *S. borealis* were also present in a cluster that had a similar configuration as in *Monilinia* and was only missing *mdpK* and one of the additional proteins. The sister species to this group of *Monilinia* plus *Sclerotinia* was not always the same. In some cases, the sister sequence in the gene tree was from a member of the *Penicillium* or *Talaromyces* genus (for instance *mdpG*, *mdpE*, *mdpD*, *mdpF*, *mdpH*, or *mdpK*), in some other cases the sister belonged to different Sordariomycetes species (such as *mdpL* or *mdpC*). In the remaining cases, the sister was uncertain, as the bootstrap value was below 50 (see *mdpA*). While we found clusters in some of the *Penicillium* and *Talaromyces* species included in the database, none of them showed similarities in terms of gene order with the cluster found in *Monilinia*. While we cannot completely exclude the possibility of HGT playing a role in the evolution of this SM gene cluster, there is no strong evidence for HGT, since multiple species across multiple taxonomic groups seem to have a variant of this gene cluster.

Another cluster that was present in *Monilinia* was homologous to the botcinic acid (BOA) cluster in *B. cinerea* (see Figure 2C). This polyketide compound is a phytotoxin that is not essential for virulence even though a *B. cinerea* double mutant lacking the BOA cluster and the cluster synthesizing botrydial was shown to be less virulent [67]. The BOA cluster consists of 13 proteins, of which 11 (BOA3 to BOA13) are found in clusters in three of the *Monilinia* species: *M. aucupariae*, *M. laxa* and *M. fructicola*. The two first genes of the botcinic acid cluster (BOA1 and BOA2) are not in the main cluster, but are found together elsewhere in the genome in *M. fructicola* and *M. laxa*, though they are not present in the genome of *M. aucupariae*. This configuration is the same one that was found in *Sclerotinia sclerotiorum*, and this species is not known to produce botcinic acid, so it is likely that the compound of this partial cluster is different. A synteny analysis of the two partial BOA clusters strongly suggested that the presence of two separate clusters (BOA1-2 and BOA3-13) is ancestral in the *Sclerotiniaceae*, whereas the joining of these clusters into a single genomic locus is a derived trait unique to the genus *Botrytis* [23]. *M. fructigena* and *M. polystroma* have lost most but not all of the genes of the BOA cluster. *M. fructigena* retains the two first genes (*Boa1* and *Boa2*) and the last one that has homologs in the other *Monilinia* species (*Boa13*), whereas *M. polystroma* retains *Boa1*, *Boa11*, and *Boa13*.

Other known pathways identified in *Monilinia* species were homologs to agnestins, prenyl xanthones, and geodin but all targeted parts of the aforementioned monodictyphenone gene cluster homolog. In addition, a homolog to the tenellin gene cluster [68] was identified, but it is unlikely that this gene cluster can produce tenellin given that the key enzyme that produces tenellin is a hybrid PKS/NRPS gene whereas the key enzyme in the *Monilinia* homologous cluster was a PKS. Additionally, the *Monilinia* cluster was missing one of the cytochrome P450 monooxygenase proteins (tenA) that catalyzes the reaction that transforms pretenellinA to pretenellinB.

We also used SMURF to detect unknown SM gene clusters (see Appendix A). The number of detected clusters per species ranges between the 14 found in *Myriosclerotinia scirpicola* and the 36 found in the outgroup *H. dingleyae*. *Botrytis* spp. have, on average, more SM biosynthetic gene clusters than *Monilinia* and *Myriosclerotinia*. *M. fructigena* has an unusually low number of SM gene clusters (16) considering the average for the other *Monilinia* species is 20. The number of key enzymes for SM biosynthesis, which refers to PKS, NRPS, DMAT, and PKS-NRPS hybrids found in the genome follows that trend as *M. fructigena* has only 22 while *M. fructicola* has 33, therefore it is unlikely that the low number of clusters is solely the result of genome fragmentation. Genes involved in the biosynthesis of antimicrobials such as solanopyrone and clavulanic acid have been identified in some *Monilinia* spp. and are usually related to antimicrobial activity [12].

### 3.6. Comparative Genomics of Sexual Reproduction and Heterokaryon Formation in Monilinia Strains

*M. laxa*, *M. fructicola*, and *M. fructigena* have been defined as heterothallic species in which each isolate carries only one of the two mating alleles (MAT1-1 and MAT1-2) [69]. Isolates possessing either of these alleles have been found in similar numbers in populations. Each idiomorph contains a pair of genes that forms the mating loci and is flanked by a pair of genes (APN2 and SLA2) that are characteristic of other heterothallic species. So, MAT1-1 contains the genes *MAT1-1-1* and *MAT1-1-5*, while MAT1-2 contains the genes *MAT1-2-1* and *MAT1-2-10* (renamed from *MAT1-2-4* as proposed by Wilken et al. [70]). A search for these mating loci in *M. aucupariae* and *M. polystroma* confirmed that the two species are also likely to be heterothallic as they only possessed one of the two idiomorphs. So, among our *Monilinia* reference sequences, *M. aucupariae* (LMK733) and *M. fructigena* (gena6) contained the MAT1-1 allele, whereas the other three species (*M. laxa* 8L, *M. fructicola* CPMC6, and *M. polystroma* SP-61) contained the MAT1-2 allele (see Appendix A and Appendix A). Beyond the reference we have used in this analysis up until now, the genomes of seven additional strains were obtained from NCBI (see Appendix A). Three of those are strains of *M. fructicola*, two of *M. fructigena*, and two of *M. laxa*. For each of these three species, genome sequences are available for isolates with each idiomorph (see Appendix A).

While signs of sexual reproduction have been observed in *M. fructicola* in the USA, analysis of European populations led to the conclusion that *M. fructicola* reproduced mainly asexually there [69]. In Australia, meanwhile, no sexual stages have been recorded yet [71]. Still, there was some evidence of recombination which could either be attributed to sexual reproduction or to the formation of heterokaryons. Heterokaryon formation is specific to filamentous fungi and is regulated by *het* (heterokaryon incompatibility) genes. Polymorphisms in *het* genes of two isolates that undergo anastomosis cause inviable heterokaryons, and consequently strains of the same species are assigned to different vegetative compatibility groups depending on whether or not they can form heterokaryons. Differences in *het* genes that cause heterokaryon incompatibility can be allelic, in which one single *het* locus contains polymorphisms that prevents the formation of a heterokaryon, or genetic, in which *het* genes found in different loci differ between strains [72]. One of the best characterized *het* loci is *het-c*. In *B. cinerea*, the *het-c* (bc-*hch*) loci was studied in a population of 117 isolates. Two different allelic types were found that could split the population in two. In addition, there was a strict correlation between the allelic *het-c* type and a resistance to the fungicide fenhexamide phenotype [73]. We searched for the bc-*hch* ortholog in the *Monilinia* species for which multiple strains were sequenced, and examined the allelic variation in the loci. In *M. fructicola*, we found that the two strains had 11 SNPs (BR32 and Mfrc123) when compared to the reference (CPMC6). On the other hand, the two strains of *M. fructigena* (Mfg5SPA and Mfrg269) did not have polymorphisms in the *het-c* loci as compared to the reference isolate (gena6). Finally, for *M. laxa*, one strain (EBR-Ba11b) had the same allelic *het-c* type as the reference (8L), whereas the other strain (Mlax316) contained two polymorphisms. Three *M. fructicola* strains carried three different *het-c* haplotypes, three *M. laxa* strains carried two haplotypes, and three *M. fructigena* strains all had the same haplotype. The higher number of SNPs found in the *het-c* loci in *M. fructicola* is in line with the general divergence found in this species where the number of SNPs per kb in the two strains was 8.7 and 4.0, respectively, while in *M. fructigena* and *M. laxa* it was less than 1 SNP/kb.

In a broader analysis, all the proteins that contained a HET domain were compared. The number of proteins with a HET domain in *Monilinia* species ranges from 19 in *M. aucupariae* to 41 in *M. polystroma*. *M. aucupariae* is the species with the lowest number of HET proteins among the species used in the phylome with the outgroup *H. dingleyae* being the species with the largest amount at 188 proteins followed by *Botrytis tulipae* at 93. Based on orthology analysis, the HET proteins in *Monilinia* were grouped into 73 different families, which do not completely overlap between different species. Among these families, one underwent a duplication that was specific to *M. polystroma* and *M. fructigena*. Comparisons of the *het* gene sequences in additional *M. laxa*, *M. fructigena*, and *M. fructicola* isolates suggested that only Mfg5SPA displayed a high degree of sequence identity and thus could belong to the same vegetative compatibility group as the *M. fructigena* reference isolate. For *M. laxa* and *M. fructicola*, the additional isolates showed significant sequence polymorphisms in *het* genes, and all are likely in different vegetative compatibility groups than the respective reference isolates.

Based on read mapping, one of the HET proteins was consistently inferred as missing. Therefore, either the gene was missing in the strains assemblies or numerous mutations affected this locus, making it impossible for reads to map there. A blast search revealed that the protein was in fact not missing but it had just accumulated enough SNPs to remain undetected by read mapping. Along with this HET protein, a second protein located next to it was also missing in all strains except for *M. fructigena* Mfg5SPA.

Comparative genomic methods have allowed us to gather information about genetic variation, differential gene expression, and evolutionary dynamics in *Monilinia* species. Comparison of whole genome sequences provides a highly detailed view of how *Monilinia* species are related to each other at the genetic level, and also provides a powerful tool for studying *Monilinia* evolution.

## 4. Conclusions

We have performed a comparative analysis of the genomes of five plant pathogenic *Monilinia* species, including the main stone fruit pathogens *M. laxa*, *M. fructicola*, and *M. fructigena*. Using a phylome approach, we analyzed the patterns of gene gain and loss in these species and found that more genes had been lost than gained; this was especially evident in *M. aucupariae*, which, despite having the largest genome, had a net loss of 973 genes. The low number of genes gained is coupled with a very low duplication rate, although the NEP1 protein family was specifically expanded in *M. aucupariae*.

Comparison of secreted proteins and the CAZome showed a low variability in *Monilinia* species. We uncovered the presence of SM gene clusters homologous to the clusters synthesizing pyriculol, monodictyphenone, botcinic acid, and tenellin. The fact that most of the clusters contain variations in the form of additional proteins or loss of parts of the cluster indicates that the final product of these clusters is uncertain. Based on previous observations, the pyriculol homologous cluster may synthesize monilidiol or chloromonilicin. This will need to be experimentally tested. The comparison of different strains of *Monilinia* species showed the presence of different mating types as previously reported for these species. In addition, we examined a wide array of *het* genes that could modulate the formation of heterokaryons. One gene in particular seemed to vastly differ between the different *Monilinia* strains, suggesting that it could be one of the main drivers of heterokaryon incompatibility. Admittedly, our comparisons are limited to a single strain per species and therefore are not informative of intra-specific variations. Some of the genes here reported as specific or lost from a given species may represent genes with variable presence-absence patterns within a species. Future studies considering genome sequences from larger, representative strain sets per species will allow the reconstruction of pan- and core-genomes per species. Genome assembly and annotation quality can influence comparative results. The used *Monilinia* assemblies had similar gene content completeness as assessed by BUSCO, and therefore we do not expect that the large differences are driven by genome quality parameters. Nevertheless, the increased use of long-read sequencing technologies will allow better coverage of telomeric and repetitive regions, which are likely to contain part of the gene variability. Finally, our results are based on bioinformatic analyses and future studies should confirm some of the predictions made.

## Figures and Tables

**Figure 1 jof-07-00464-f001:**
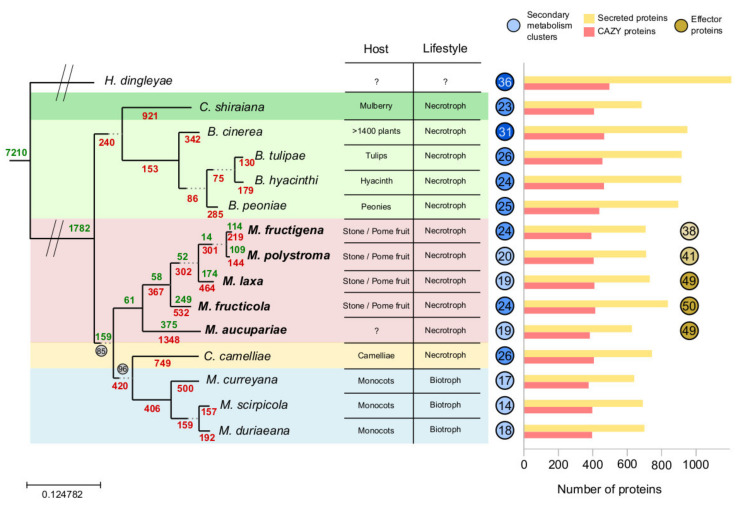
The tree represents the evolution of the species included in the phylome. Support is 100 for all nodes except the two nodes where a number can be found surrounded by a circle. Green numbers placed in the tree indicate the average number of genes gained at that particular node, whereas red values indicate the average number of losses. The table placed next to the tree indicates, for each species, the preferred host and the kind of lifestyle. The number within the blue circles indicate the number of secondary metabolism gene clusters detected by SMURF, while the numbers in the brown circles indicate the number of effector proteins found in each *Monilinia* species. Finally, the bar plot indicates in yellow the number of secreted proteins in each genome and in yellow the size of the CAZome.

**Figure 2 jof-07-00464-f002:**
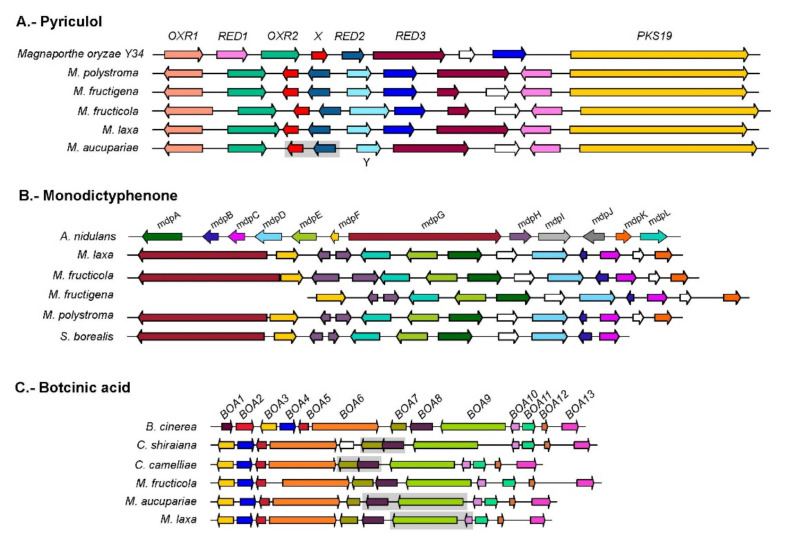
Schematic representation of the three main secondary metabolism gene clusters discussed. Arrows represent genes and colors denote homologous relationships within each cluster. Two genes surrounded by a grey area indicate the genes were predicted as a single gene.

## Data Availability

All phylogenetic trees reconstructed in the phylome can be obtained from phylomedb.org. All additional data is available upon request.

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
