# Peer review of "Comparative Genomics Used to Predict Virulence Factors and Metabolic Genes among Monilinia Species"

_jof, 2021, doi:10.3390/jof7060464_

Round 1

Reviewer 1 Report

This manuscript describes the comparative analyses of the genomes of several species of Monilinia which are known fungal pathogens of plants. Overall, the manuscript is well written and clear. The introduction nicely lays out the importance and rationale for the work and the analyses appear to have been done properly using several methods and software packages that are standard to the field of fungal genomics. The merit of this work lies in the supplemental tables which will allow researchers interested in studying these species to determine the prevalence of their gene of interest in the isolates examined in this work. My major criticism of the work is that the authors place entirely too much emphasis on species-specific gene content, gene loss and gene gain based on the genome of a single isolate from each species. There is no discussion about intra-species variation. This critique can be addressed by sequencing another isolate or two of the 5 main Molinia species discussed and redoing the analysis. This would significantly improve the confidence in the ability of the authors to determine the species-specificity of a gene.  Furthermore, the genome assemblies for each of the isolates (as displayed in Table S1) are highly fragmented genomes and therefore it is possible that some of the genes that are thought to be absent from a particular isolate might reside in a region of the genome that was unable to be included in the final assembly.

Author Response

We appreciate the suggestions of the reviewer that it will be great to sequence more than one strain for each species. The assemblies of some of the strains were not developed in this work and we just used the ones deposited in NCBI. Improving existing assemblies or characterizing the genomic diversity within a species was not the main goal of this work. These analyses would require access to representative collections of strains from different locations, pre-screening with some genetic markers to ensure proper representation of the existing genetic diversity, and re-sequencing of a sizable number of strains to establish the pan- and core- genomes from these species. We consider that doing this is out of the scope of our study. We have, however, discussed the limitation of considering a single strain per species, and that our results cannot inform on the within species variation.

Reviewer 2 Report

In this study, the Authors performed a comparative analysis of the genomes of five plant pathogenic  Monilinia species, including M. laxa, M. fructicola and M. fructigena,  using a phylome approach.  This study is very interesting and constitutes a fundamental approach to understand the genetic mechanisms of the fitness and pathogenicity of these necrotrophyic plant pathogens.

The paper is well and clearly written and in my opinion deserve to be published without any substantial change. See notes in the text (attached file) for very minor suggestions.

Author Response

We appreciate all suggestions of the reviewer. We have modified and corrected all the suggestions made by the reviewer (highlighter in yellow)

Reviewer 3 Report

Summary

Marcet-Houben et al. present a well-written manuscript that entails a broad comparative genomics study on Monilinia species. By retrieving publicly available sequencing data from multiple Monilnia species they performed phylome analysis to determine CAZome, virulence-associated genes, secondary metabolism clusters. Furthermore, they provided a detailed projection of some of the identified gene clusters and compares these to related model species. The bioinformatics performed in this manuscript is well performed. I have some Major comments to address and some minor ones:

Major:

  • Title suggests the use of functional prediction of proteins,
    “Comparative genomics used to predict virulence factors and metabolic gene clusters among Monilinia
    would reflect the content of the manuscript better.

  • The abstract is not very well written and does not reflect the quality and content of the manuscript. Please rewrite the abstract. For example, remove (line 30-31) “rate of spread in different geographical areas” as this is not mentioned in the manuscript.

  • Because of the use of many strains, it is often hard to keep track of what is performed with which strain. Please make it clearer.

  • Please give an estimation of the genome completeness by using either BUSCO or CheckM. aucupariae results need additional quality assessment to confirm its more distinct genome content and its high number of gene loss.

  • A lot of analyses are performed. The manuscript would aid from providing the reader with an overview of all the analysis performed in a workflow.

  • The results and discussion section has limited discussion on the bioinformatic limitations and strongpoints. As the authors uses many bioinformatic approaches, more thorough discussion would make the manuscript even stronger and would help the community to learn from performing these kind of genome comparisons.

Minor:

  • Table S1, please add bioproject/SRA code or Accession number for all strains.
  • Table S1, please highlight or note the two sets of strains to better identify which strains are used for which analysis.
  • Line 98-99, “The version described .. RKRL01000000.” Please add this information to Table S1.
  • Line 101-102, Remove “The raw PacBio .. SRR8146337.” This information is identical to Vilanova et al. 2021 (PMC8070815) and suggest that it is a new deposited sequence.
  • Line 109-110, “To reconstruct … were collected (see Table S1).” Please clarify what is meant with collection of evolutionary histories? Suggestion, Data from 15 different species of the class Leotiomycetes were collected.
  • Line 112-114, The authors use PhylomeDB and homologous proteins are determined using BlastP, reciprocal best hits (RBH) is one of the standards to determine these hits. Can the author clarify why they use unidirectional blast?
  • Line 126-127, IDs 383 to 387 are not available via phylomedb.
  • Line 131-132, please define which blast algorithm was used.
  • Line 156, please add version number of DupTree.
  • Line 169, 178 and 181 what blast algorithm was used.
  • Line 198-199, “The SRA project for a strain …” please remove sentence as does not matter that much that it is not used.
  • Line 199, was this performed using BWA or BWA mem?
  • Line 199-203, This is very confusing, why are the reads not covering its own reference template? Could authors please clarify because identifying genes as being absent based on reference mapping and not based on reference genome content could greatly affect the outcome.
  • Line 218-220. Can authors please add the individual trees to the supplement.
  • Line 237-238, Authors mention the use of GO enrichment, if so please add to the method section how this was performed.
  • Line 241-242, Here authors mention that some hits were found in Leotiomycetes not included in the phylome. Why are these not included in the phylome analysis? Are these individual sequenced genes and not of a full genome?
  • Line 264-267, multiple mentions of BcNEP, please use gene name and write italic.
  • Line 281, “.. smallest scretome aucupariae” See major comment above, additional quality estimates need to be given to ensure that the genome is of good quality.
  • Figure 2, A-Pyriculol, please write genes in italic.
  • Figure 2, B-Monodictyphenone, please adjust color (light gray) of gene mdpI, currently barely distinguishable from non-related genes (white).
  • Line 358-374, it is unclear if the RNA-Seq analysis was reperformed in this manuscript or that it is reused from another study. If performed in this study please add it to the method. Otherwise
  • Line 361, change RNA Seq to RNA-Seq
  • Line 378, “most of the cluster genes were present” could authors please be more precise on the definition used.
  • Line 383, change Figure 2 to Figure 2B
  • Line 386-389, please write genes italic.
  • Line 388-392, I agree with the explanation of the breakup of the cluster. Again, I would like to have some additional quality measures of the assemblies. This would give the analysis more strength.
  • Paragraph 3.5, please write all genes in italic.
  • Figure S4, the caption does not match the figure.
  • Figure S4, why does the mdpG tree only have three Monilinia species in the tree and not four?
  • Line 402-403, This can be seen from Figure S4, please indicate A.  nidulans.
  • Line 437-438, please write genes italic.
  • Line 466-474, it’s a bit of a puzzle to interpret Table S1 to see all the mating types. It is correct; however, I would suggest adding the strain names to this paragraph. This is also more accurate.
  • Line 481, here het is written italic, later it is written in caps(HET). Please adjust accordingly.
  • Line 515-525, This is indeed a good point that the authors make. The manuscript would aid in adding a broader discussion on how to solve these comparative limitations. I can imagine with more and more applicability of PacBio or Oxford Nanopore sequencing we will be able to obtain near-complete genomes which would limit the number of difficult to identify regions.
  • Line 531-532, This conclusion can only be drawn with respect to the quality of the genomes used.

Author Response

We would like to thank the referee for all the suggestions made, which have been answered one by one in the attached file and introduced in the new version (highlighted in yellow)..

Round 2

Reviewer 1 Report

The authors have adequately addressed my concerns.